# Fish Loss/Waste and Low-Value Fish Challenges: State of Art, Advances, and Perspectives

**DOI:** 10.3390/foods10112725

**Published:** 2021-11-07

**Authors:** Angela Racioppo, Barbara Speranza, Daniela Campaniello, Milena Sinigaglia, Maria Rosaria Corbo, Antonio Bevilacqua

**Affiliations:** Department of Agriculture, Food, Natural Resources and Engineering, University of Foggia, 71122 Foggia, Italy; angela.racioppo@unifg.it (A.R.); barbara.speranza@unifg.it (B.S.); daniela.campaniello@unifg.it (D.C.); milena.sinigaglia@unifg.it (M.S.); mariarosaria.corbo@unifg.it (M.R.C.)

**Keywords:** perspectives, sustainability, loss and waste, unwanted fish, global solutions

## Abstract

The sustainability of fishery is a global challenge due to overfishing and reduced stocks all over the world; one of the leading factors of this threat is fish loss/waste. As a contribution to the global efforts towards a sustainable world, this review addresses the topic from different sides and proposes an overview of biorefinery approaches by discussing bioactive compounds that could be produced from fish loss (nitrogen compounds, lipids, minerals and pigments, and fish-based compounds such as chitosan). The second part of this review reports on the possibility of using loss or unwanted fish to design products for human consumption or for animal feeding, with a focus on economic criteria, consumers’ segmentation, and some examples of products. The final focus is on Food and Agriculture Organization FAO guidelines as a roadmap for the future with respect to solving this threat by addressing the problem from different sides (technology, skills, market, policy, social and gender equity, and infrastructures).

## 1. Introduction

Food sustainability must be a primary goal worldwide if we want to preserve our planet for the future generations; however, this ambitious goal is a challenge because there are two critical issues in food handling, production, and storage: food loss and food waste. The magnitude of these challenges is so important that the United Nations prescribed Target 12.3 in Agenda 2030 and defined the following Sustainable Development Goal: by 2030, global food waste at the retail level must be halved and food losses along the entire production chain must be reduced [1,2].

Fish is a perishable raw material and, more than other matrices, generally experiences the challenges of food waste and food loss (up to 40% as physical loss in some middle-low income countries), although this is a paradox because the consumption of fish is below the recommended levels worldwide [3]. FAO estimated that of the 178.5 million tons (MT) of seafood, mainly (at least 88%) intended for human use, of the total amount of seafood, 96.4 MT were from fish capture (mainly anchoveta, Alaska pollock, skipjack tuna, herring, and blue whiting) and 67.1 MT from farmed species (shrimp, prawns, salmon, molluscs, tilapia, catfish, sea bass, and sea bream) [4,5]. The projection for fishery development predicts a further increase up to 200 MT in ten years, while the amount of sustainable fishery is significantly decreasing (ca. 66% in 2017 versus 90% in 1990) [4].

An important initiative is Food Lost and Waste in Fish Value Chains [6]. It is an important tool because it focuses on both scenarios (causes and economic value) and on solutions (both from the side of consumers and producers).

A global challenge requires holistic and global solutions, as this threat was studied in the past from different point of views (biorefinery, designing new foods, and aquaculture feeding). This paper offers a possible holistic insight, at least from the side of food technologists and food engineers, and addresses some possible scenarios: biorefinery and the design of new products for human consumption or animal/aquaculture feeding (Figure 1).

After an insight on the definition, the paper is organized as follows: biorefinery (Section 3 and Section 4), design of new foods for human consumption (Section 5), and the use of ingredients and low-value fish for animal feeding/aquaculture (Section 6). As a final step, future methods are highlighted by keeping in mind that global problems require holistic and global solutions.

## 2. Food Loss and Food in Fishery: Definitions and Why They Are a Challenge

Although used in an interchangeable manner, the concepts of food waste and food loss are different; thus, FAO defined them to help policy makers and stakeholders in order make appropriate decisions. Food loss was defined as a decrease in the quantity and quality of food because of the decisions made by food suppliers, excluding retail, providers, and consumers [7]. The best example of food loss in fishery is the so called low-value fish, which is generally discarded because it is not regarded as of sufficient quality for consumers. On the other hand, food waste is the decrease in quantity or quality of food because of decisions made by consumers, providers, and retail [3]. This distinction clarifies that food waste is the result of actions taken by consumers or on behalf of consumers, while food loss depends on the actors of supply chain but not by consumers. There are other terms that are important in light of sustainable fishery and its many stakeholders; a synopsis is provided in Table 1.

Generally, food loss and waste are referred as “loss” because they share several causes and are both produced along the entire chain from production to consumption. Kruijssen et al. [3] described four kinds of losses (physical, quality, nutritional, or market force loss) and studied their causes; thus, they highlighted five possible phases where loss and waste occur: primary production, post-production, processing, distribution, and consumption.

As reported by the authors [3], losses could occur during primary production, which for fishery means capture or harvest (for fish farming), due to failing from nets or injuries when fishermen remove catches from nets because fish remains in the nets for a long time after the capture or from the use of harmful techniques during capture or farming and the lack of chilling.

The second step for loss and waste is post-production (landing, handling, and storage, logistic) mainly due to inappropriate conditions of storage, transportation, delays in sale, and infestation by parasites [3].

During processing (gutting, drying, fermentation, canning, filleting, and packaging), the causes of loss and waste include the use of inappropriate packaging, poor quality water used during this phase, low processing capacity, and infestation/predation by insects, birds, and rodents. During distribution (retail and transport), the causes generally rely on delays in packaging or transport (for example, because landing sites are too remote) and an excess of supply or a careless handling, whereas for the last step (consumption: storage, preparation, and table), the main causes are spoilage, excess in preparation, and discards [3].

## 3. Biorefinery: Definition, and Classification

Zero Waste is a comparatively new concept based on the idea of considering any waste material produced by human activities as a possible resource for other processes or users. The aim is for no trash to be sent to landfills, incinerators, or the ocean. The production and the disposal of waste are serious problems at a global level. According to Eurostat [9], 483 kg of municipal waste per capita was generated in the European Union EU in 2019, and only 45% of those wastes were recycled and composted (combined) during that same year.

Agricultural, forestry, and fishing activities alone produce about 40 million tons of waste per year in Europe. The biological origin of these wastes makes them an interesting raw material for recovering natural molecules of industrial interest or to obtain other value-added molecules, biomaterials, and biofuels. These transformations can be carried out through the application of biotechnological processes.

In general, biorefineries can be intended for the recovery of natural molecules (biomolecules) of industrial interest or for the bioconversion of organic material. Traditional petrochemical refineries differ from biorefineries in terms of the raw material used and the final products. In a refinery, fossil resources are converted into energy and chemical compounds, whilst in a biorefinery, bio-resources are used to be converted into useful compounds [10,11].

Due to the wide variety of residual biomass obtained as waste from agro-industrial activities, the possibilities of enhancing organic matrices by using biotechnological approaches are manifold.

The biorefinery concept dates back to the beginning of the industrial era. Nowadays, due to the rise in oil price and the concern climate changes, biorefineries and bio-based products are viewed with great interest.

In 2012, the International Energy Agency (IEA) Bioenergy Task 42 [12] developed a classification system for biorefineries based on the following: 1. biorefining platforms or key intermediate products and processes; 2. final products: energy (biofuels, power, and heat) and material products (chemicals, building blocks, food, and feed); 3. feedstock: crops from agriculture (e.g., starch crops and short rotation forestry), residual biomass coming from agriculture (e.g., straw and cattle manure), forestry (e.g., bark), or industry (e.g., used cooking oils and waste stream from biomass processing); 4. the process used to convert biomass: biochemical (e.g., fermentation and enzymatic conversion), thermochemical (e.g., gasification, pyrolysis, and combustion), chemical (e.g., acid hydrolysis, synthesis, and esterification), and mechanical (e.g., fractionation, pressing, and milling) [12,13]. These processes are complemented by hybrid conversion platforms, which combine the thermochemical pre-treatment phase and a biological conversion phase [14,15].

Kamm and Kamm [10] and Clark and Deswarte [16] classified biorefineries into three types: (1) phase I biorefinery: it uses only one feedstock material, has fixed processing capability, and produces a single primary product (biodiesel from vegetable oil, pulp, and paper mills and the production of ethanol from corn grain); (2) phase II biorefinery: it uses only one feedstock, but it can produce various products; (3) phase III biorefinery: can use different types of raw materials, processing technologies, and produce more types of products.

Kamm et al. [17] proposed another type of classification based only on the type of feedstock used: (i) Lignocellulosic Feedstock Biorefinery; (ii) Whole Crop Biorefinery; (iii) Green Biorefinery; (iv) Organic Waste Biorefinery (Table 2).

## 4. Fish Based Bioactive Compounds

Each year, discards from global fisheries exceed 20 million tonnes, which is equivalent to 25% of total marine catch production; discards measure up to 5.2 million tonnes per year in the European Union [18]. Al Khawli et al. [19] estimated that the number of fish by-products represents 30 to 85% of the weight of the different catches. These by-products are predominantly composed of heads, viscera, skin, scales, bone, cut offs, backbone, and blood. Another source of potential fish by-products includes low-value fish and so-called “unwanted” fishing [20].

Usually, fish waste is burnt, landfilled, dumped in the sea, or even abandoned, with a range of negative impacts on human health, biodiversity, and the environment. Therefore, over the years, the need to transform this waste into new resources in an environmentally sustainable manner has become increasingly important.

Moreover, several bioactive compounds such as protein, fatty acids, proteins, vitamins, minerals, and other fish by-products can be extracted and reused not only for the food sector, but also for pharmaceutical industries, cosmetic industries, or for biofuel production [20].

From an economic point of view, the value of fish waste and loss is gradually increasing. In the last report of European Market Observatory for Fishery and Aquaculture EUMOFA [21], there is a technical definition for fish waste and loss as Rest Raw Material (RRM) that comprises all the potentially useful material removed from fish, shellfish, crustacea, and other species to prepare biomass for food or no-food uses.

An idea of its economic value could be based on the projection of the World Bank model, which estimates that, in the near future, 15% of fishmeal will be derived from RRM [21].

The state of art and the current perspectives are greatly variable, at least in Europe, depending on the infrastructures and the willingness of consumers to pay for such products. In Europe, the biorefinery approach is becoming more and more important, as inferred by the number of factories able to act in the pathways G (two-platform oil and biogas biorefinery using aquatic biomass) and K (One-platform bio-crude biorefinery using lignocellulosic, aquatic biomass, and organic residues), with two large companies acting on aquatic biomass and the potentialities of upgrading their main activity towards other wastes [22]. In other countries, the Biorefinery Outlook lists three main companies situated in USA, Australia, and Canada [22].

From a global point of view, the impact of biorefinery, including fishery biorefinery, is expected to experience a strong increase because the last report indicates at least 400 companies/factories involved in this new trend/philosophy in EU [21].

The goal of this section is to provide an overview of what can be produced or extracted from fishing industry waste and the pathways for their valorisation.

### 4.1. Nitrogen Compounds

The protein fraction of fish is a valuable nutrient source for both humans and animals, especially because of its complete amino acid profile. Fish proteins have higher nutritional values than vegetable proteins and are found in large quantities in bones, head, viscera, liver, kidney, eggs, and skin from which they could be extracted by enzymatic or biological hydrolysis.

Several therapeutic properties of bioactive peptides have been demonstrated, including antihypertensive, antioxidant, antimicrobials, and antiproliferative effects [23,24]; immunomodulators [25]; and anti-hyaluronidase and anti-tyrosinase effects [26].

Protein hydrolysates generated from fish proteins are good food supplements because they contain bioactive compounds that can be easily absorbed and used for various metabolic activities. In the food industry, they have been successfully incorporated into various cereal, meat, fish, and cracker products.

Many countries use fish protein hydrolysates as functional foods and/or nutraceuticals. Some products currently on the market include the following: Vasotensin^®^, a supplement based on peptides derived from fish bonito waste with hypertensive effects; Seacure^®^, a dietary supplement formulated from white fish filleting waste to promote gastrointestinal health; Fortidium Liquamen^®^, another dietary supplement based on protein hydrolysates from fish guts with anti-stress and antioxidant effects; Peptydiss^®^ a supplement based on hydrolysed proteins from sardines with anti-stress and sleep-disrupting effects; Stabilium^®^ 200, useful for supporting intellectual faculties and memory; Nutripeptin^®^, a supplement derived from cod with an antidiabetic effect [27].

Moreover, hydrolysed proteins extracted from fish by-products can also be used for animal feed. Indeed, these compounds have been shown to be nutritious feed ingredients and potential substitutes for fishmeal in aquaculture diets [28]. Therefore, they could become a viable alternative to herrings and anchovies and are usually used as a protein source for fish feed production. Seafood proteins also possess important and unique technical-functional properties such as water-binding, emulsifying, film-forming, foaming, and gel-forming capacities [29]. Due to these characteristics, proteins extracted from fish waste can be used for industrial bioplastic production usable as new packaging materials.

Uranga et al. [30] and Araújo et al. [31] used fish by-products in the form of fish gelatine or myofibrillar proteins in order to develop a low-cost biofilm, which can easily incorporate additional substances such as anthocyanin, as well as chitosan, extracted from fish waste. The resulting biofilm prevented the oxidation of lipids in packaged food and, thus, improved its shelf life.

The antioxidant activity of some bioactive peptides extracted from fish waste can have a positive effect on some clinical conditions, such as inflammation, cancer, cardiovascular problems, atherosclerosis, and, more generally, the aging process [32].

The use of enzymes extracted from fish waste is very interesting for several industrial processes (e.g., food processing, cosmetics, and textiles). Pepsin, for example, is used for collagen extraction and as a rennet substitute in dairy production [33]. Collagen is contained in large quantities in fish skin, from which it is extracted by acid or basic hydrolysis, and is widely used in the pharmaceutical and cosmetic industries and as a food supplement. Marine collagens can be obtained from several sources: sponges, jellyfish, and fish offal such as bones, skin, scales, and fins. In biomedical and pharmaceutical sectors, collagen has several applications: it is used as an excipient for drugs and in the biomedical field to produce human skin substitutes, blood vessels, and ligaments. Several studies evaluated its antimicrobial, antioxidant, and antihypertensive properties [34,35]. Gelatine, which is obtained by the partial hydrolysis of collagen, is used as a gelling agent in food, pharmaceuticals, and cosmetics and has been shown to have antioxidant and antihypertensive properties. Other enzyme derivatives are protein peptones, a mixture of polypeptides and amino acids obtained from the controlled enzymatic degradation of proteins. They are used to produce culture media for microbiological or biotechnological purposes and are generally the most expensive ingredients [36].

### 4.2. Chitin and Chitosan

After cellulose, chitin is the second most abundant natural polymer on earth. It can be obtained from various sources and is mainly found in the exoskeletons of arthropods (insects, crustaceans, and arachnids) and molluscs. It is not strictly a fish bioactive compounds, but it was included in this section because it could be classified as a fish-based compound and is one of the most important product of fishery biorefinery.

A derivative of chitin is chitosan produced by the partial deacetylation of chitin by chemical or biological methods. Chitin biomolecules and its derivatives (chitosan, chito-oligosaccharides, and glucosamine) have excellent biodegradability and have been shown to have numerous biological properties (antimicrobial, antitumour, anticoagulant, antioxidant, antimutagenic, and cholesterol-lowering). They also find various industrial applications in agri-food, textile, pharmaceutical, and cosmetic sectors.

Chitosan has good antimicrobial activity and antioxidant effects that effectively prevent the oxidation and rancidity of food lipids and inhibits the growth of microorganisms; thus, it is used as a food preservative to extend the shelf life of food. Currently, several dietary supplements, dietary fibres, and nutraceutical products based on chitin and chitosan are commercially available [37,38].

Furthermore, chitosan and its derivatives have excellent film-forming properties and antibacterial properties, making them useful in food packaging and preservation [39].

Bhuimbar et al. [40] developed an antibacterial active food packaging film by using a preparation of collagen-chitosan extracted from fish waste. Film showed antibacterial effects against several food pathogens such as *Bacillus saprophyticus*, *Bacillus subtilis*, *Salmonella* Typhi, and *Escherichia coli*.

In the food industry, chitosan and its derivatives are also used as food additives such as thickeners, decolorants, and stabilizers [41] or as a source of dietary fibre in value-added food products (e.g., ingredient in food supplements).

These molecules are currently used in the environmental industry for water treatment, heavy metal removal, or in drinking water processes. Chitosan has flocculating, chelating and adsorbent properties, which are useful characteristics for industrial wastewater treatment [39]. Several studies have shown that chitosan and two water-soluble chitosan derivatives (N-N-N-triethylammonium chitosan and carboxymethyl chitosan) were able to remove heavy metal ions in contaminated water, thus proving to be a viable method for wastewater treatment [17].

### 4.3. Lipid Compounds

Fish wastes, especially heads, skins, and viscera, are an important source of fatty acids from which oil can be extracted for both human consumption and biodiesel production. Fish oil is rich in omega-3-fatty acids, such as eicosapentaenoic acid (EPA) and docosahexaenoic acid (DHA) derived from α-linolenic acid (ALA). The therapeutic importance of omega-3 for the prevention of cardiovascular diseases, hypertension, and for asthma as well as attention deficit/hyperactivity disorder in children is well known. Other benefits include the anticoagulant effect, preventing dementia, anti-inflammatory properties, and as an antidepressant. Furthermore, fish oil is also a good source of vitamins (E, D, and A) and squalene. Vitamin D_3_ supplementation, extracted from fish oil, has been shown to be potentially useful in the treatment of atopic dermatitis, ulcerative colitis, and Crohn’s disease and is generally recommended as a preventive measure in all cases where vitamin D_3_ deficiency needs to be prevented (e.g., premature infants, lactating and menopausal women, the elderly, in cases of individuals with low or no sun exposure, long-term treatment with anticonvulsants or corticosteroids, and people with low blood pressure) [42].

Squalene is extracted from the liver, stomach, pancreas, kidneys, and other organs of some shark species. However, Europe has drastically reduced shark fishing quotas in recent years in order to protect the marine ecosystem. It is, therefore, crucial to find alternative sources of this valuable biomolecule. Squalene finds application in the cosmetic, food, and pharmaceutical industries. Its potential anticancer, anti-fungal, antioxidant, and antibacterial properties are widely studied [38]. Finally, squalene can be introduced into the human diet as a supplement to support cardiovascular and joint function [38].

### 4.4. Pigments and Minerals

Fish waste is also a source of natural pigments, such as carotenoids, and minerals. The most commonly found carotenoids are lutein, β-carotene, α-doradexanthin and β-doradexanthin, zeaxanthin, canthaxanthin, and astaxanthin. Astaxanthin (3,3-dihydroxy-β, β-carotene-4,4-dione) is the main carotenoid in marine and freshwater fish (e.g., salmon and sea bream) and accounts for 74–98% of the total pigments in crustacean shells (e.g., lobster, shrimps, and crabs). It is widely used in food and pharmaceutical industries as a precursor of dyes, antioxidants, and vitamin A, and some studies showed its potential anti-cancer and immunostimulant activities [38].

The bones are an important source of inorganic minerals, such as hydroxyapatite, calcium, phosphorus, zinc, and iron, which can be used as food supplements. Hydroxyapatite extracted from natural sources presents excellent bioactivity, osteoconductivity, and osteoinductivity and, in general, has better characteristics as a biomaterial than synthetic and finds wide usage in medical or dental applications [43,44].

## 5. Low Value Fish or Unwanted Catches: Criteria and Idea to Design New Products

Apart from food waste and loss and the possible utilization of by-products, as described in the previous sections, there are two challenges in fishery that require appropriate decisions and solutions from stakeholders, policy makers, and consumers: bycatch/unwanted catches and low value fish species.

Although different, these challenges could have common solutions, as briefly reported in this section. In 2013, the European Commission introduced the Landing Obligation (LO) or “discard ban,” which stated that all catches of species subject to catch quotas and/or minimum conservation reference size (MCRS) must be landed and will be counted against the quota [45]. Although this measure has been implemented, there is still a significant quota of unwanted catches or by-products, which requires alternative solutions. Connected to this threat, there is a need for decisions to made with respect to low-value fish.

The category of low-value fish is complex and is composed of at least three different groups of products/species [46]: (i) catches discarded for their morphological characteristics (low body growth and edible part not attractive for consumers); (ii) accidental catches, which are generally refused for consumer habits and ancient tradition; and (iii) fishes of moderate value in which its sale could be not profitable.

The University of Wageningen has developed an online dossier to address the challenge of discards and unwanted catches with regulatory framework, definition, new research topics, and possible solutions [47]. This dossier is currently updated and improved with the most recent advances in research and technological transfer.

The Waste Framework Directive of EU in 2008 established a hierarchy of solutions, also shared by the US Environmental Protection Agency [45]. The most important solutions in this list include prevention and reduction and then use of loss/waste for new products intended for human consumption.

This second choice is very important in fishery, considering that fish proteins generally covers 17% of protein intake [48], and there is an increasing demand of fish due to healthy, social, and economic reasons.

When designing new food, the first challenge is to point out the target: the consumers for which the product is intended to. In the field of fishery, Silva et al. [49] and Corallo et al. [50] reported four possible targets of consumers: (a) Individualist—food trends depend on personal reason (mainly economic considerations, habits, and mood); (b) Foodie—food choice relies upon sensory and organoleptic properties (flavour, texture, general appearance, freshness, etc.); (c) Health enthusiast—food choices depending on label, claims, and nutritional or functional properties; (d) Environmentalist—food choices based on sustainability.

The use of discards and low-value fish could catch the attention of environmentalist consumers, but if some factors/variables are stressed in a good manner (for example, by using interesting technological formulations or by pointing out positive health effects), foodie and health consumers could be caught too.

The second challenge is the technological goal; for the purpose of low-value fish and by-products/discards, a possible goal could be referred to as reformulation [49] because raw materials generally require some approaches intended to improve sensory scores, to prolong the shelf life through preserving treatments, and to increase the economic value by other factors (for example by adding a convenience value). Reformulation is generally the result of a combined action of designers/innovators in the lab and experts in economy and food choices who try to know consumer habits and preferences in order to develop or to reformulate a product able to satisfy their needs [51]. This process generally starts with a survey about consumers’ diet and choices and aims to gain knowledge on consumer profiles (gender, age, and occupation) in order to identify a potential target market segment [52].

Finally, the last challenge is to elucidate the reason beyond all these processes, as pointing out the reason is a key factor for the promotion of the product and for a correct definition of price, targets, places, and supply chain. Figure 2 shows a graphical overview of the factors to address when designing new food.

This approach or the design of new products from low-value or discards has been tested since the mid 1900s in the categories of processed and ready-to-eat food; for example, fish pulp was the basis to produce surimi or other restructured products or derivatives [52].

After the lab phase and the optimization of the formulation, before effective scaling up, there is an economic matrix to fill in, and there are decisions made on the opportunity of industrial validation.

The approach was developed as a deliverable in project H2020 DiscardLess [53]; it is based on some criteria and scores, and a summary is provided in Table 3; some products fail because they do not meet the criteria for TRL (technology readiness level), costs (production process has costs that are not compatible with economic valorisation), or availability of facilities.

There are thousands of solutions and possible approaches depending also on the habits and tradition of different countries; hereby, we focus only on some examples that could be regarded as best practices, as reported in Table 4.

## 6. Waste or Unwanted Catches for Animal Feeding

The cascade of biorefinery indicates another high-value pathway for the utilization of fish waste and loss: animal feeding. A survey published in 2021 reports at least 16 studies conducted all around the world on this topic (Brazil, China, Iraq, Japan, Saudi Arabia, Lebanon, and Mediterranean countries of Africa), with an increasing trend in the last two years [61].

The most important part for fishery wastes is the use of active compounds/components, as extracted by the biorefinery approach reported in Section 4, to produce pellets for aquaculture. Generally, the components or fishmeal represent a part of a formula (around 10%) in combination with fruit/vegetable, cereals, starch, and other ingredients [62,63]. The supplementation with some active compounds from fishery (vitamins, proteins, and fatty) has some benefits in terms of enhanced digestibility of formula, increased FCR (feed conversion ratio), higher weight gain, and protein efficiency ratio (PER) [64]; an example of the practical benefit of this approach was reported by Mo et al. [65], who described increased protein digestibility in grass carp and tilapia from 65 to 80% when the feed was improved with a combination of vitamins and minerals.

A cheaper approach is the use of by-products (heads, viscera, skin, and skeleton), treated with enzymes, to achieve a protein hydrolysate [66] used as feeding formula for salmon, shrimp, rainbow trout, and cod [67].

A survey in the literature and a recent review [68] suggests some possible methods for using wastes and unwanted catches for animal feeding, particularly for aquaculture:
Skin, biomass, and bones as sources of gelatin, proteins, and calcium;Homogenate and meal from small fish (croaker, horse mackerel, flying fish, chub mackerel, and sardine) as the main ingredients of fish formula and pellet for high-value species;Homogenates from shrimp waste (heads, appendages, and exoskeleton) as a source of lysine;Silage and fermentation with lactic acid bacteria to produce homogenates with a higher shelf life and/or biological value.


Finally, another new method is the use of a codesign food system and a circular approach; in this context, fish loss and waste are not directly used as feed for aquaculture but as feed for black soldier fly larvae, which in turn are protein sources for aquaculture. There are several benefits for this approach, including the high conversion rate of by-products into insect biomass and the high value of proteins [69].

## 7. Social Impact and Guidelines for the Future

The challenge of loss in fishery is a global problem and should be addressed through different directions/lines. This paper addresses the technological point of view, with a focus also on the economic (consumers’ segmentation and criteria to design a new product) and chemical sides (biorefinery); however, a global threat requires a multilevel solution that is able to propose new methods from different point of views.

The reduction in loss and waste in fishery, as well as in all food chains, can be overcome only if the leading factors are removed by adequate technological, political, and economic decisions. In this context, FAO [2,4] proposed a multi-tasking method able to counteract the problem at different levels:(a)The improvement of the technological tools in terms of fish capture, cold chain, and quality of water used for processing in order to avoid unwanted capture, ghost fishing, or discards due to spoilage;(b)Efficient infrastructure in terms of roads for logistic and adequate plants for processing;(c)Education and lifelong learning to improve skills of fishermen and all actors of the chain;(d)A market driven innovation to meet the requirements of consumers and to design new foods or products.

In addition, this multitasking solution should also consider an adequate regulatory framework, which is lacking in some countries, as well as appropriate actions by policy makers to remove the barriers (gender or social hurdles) responsible for inefficient processing (Figure 3).

From a technological point of view, biorefineries and the design of new products for human or animal consumption are promising methods, although some challenges should be addressed such as scaling up some technologies, the costs, and the ability to capture a market quota.

## Figures and Tables

**Figure 1 foods-10-02725-f001:**
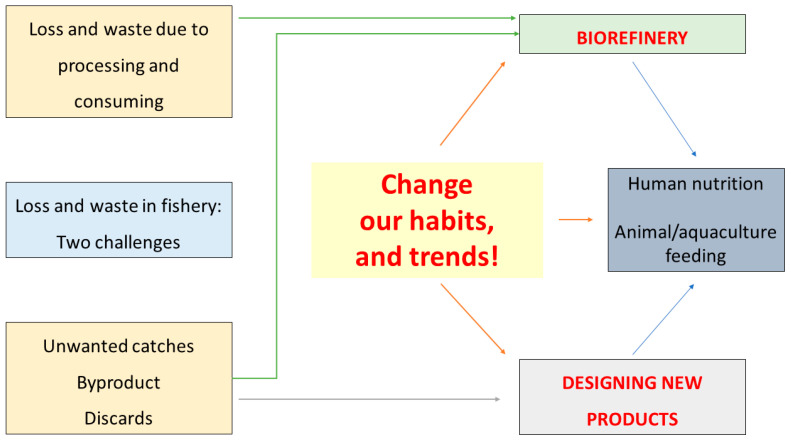
The figure describes the two threats beyond this paper along with the possible methods (biorefinery and production of new foods for human consumption and animal feeding).

**Figure 2 foods-10-02725-f002:**
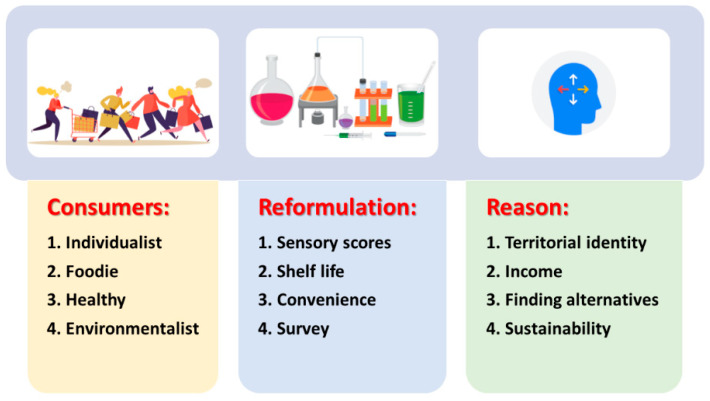
The design of new foods from discards or loss in fishery should rely on three main variables: the target (consumers), the technological aim (reformulation), and the reasons beyond the entire process; this scheme can be regarded as the “virtuous cycle” for new product design from low-value or discards of fishery.

**Figure 3 foods-10-02725-f003:**
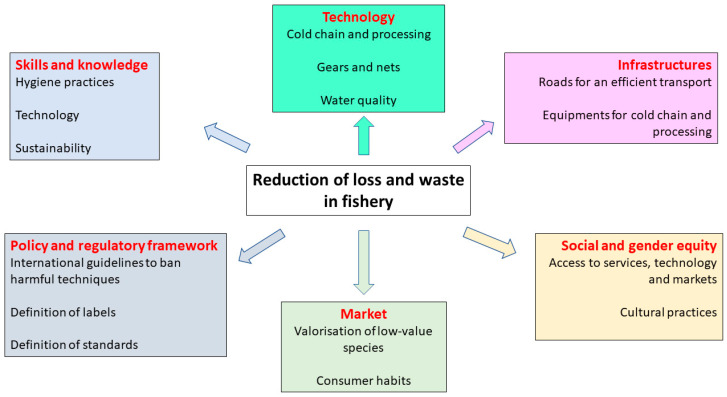
The challenges of food loss and unwanted catches require solutions in at least six different lines. This figure reports on the needs to be addressed for each line.

**Table 1 foods-10-02725-t001:** A glossary for a better understanding of food waste and loss in fishery. The definitions and the concepts have been found and modified from Goodfish [8].

Term	Definition
Aquaculture/Fish farming	Fish or plant farming for food
Broodstock	Fish used for providing eggs or larvae; they may be sourced from wild or grown in captivity
Bycatch	Aquatic life killed or damaged during fishing but not retained for being sold. It is a kind of unwanted or accidental injury to wildlife
By-product	Any fish or shellfish retained in fishery before being sold but not sought by fishery itself
Capture fisheries	Wild aquatic life caught for being sold
Closed or recirculating system	Aquaculture system that usually recycles most or all water
Discarding	Returning unwanted catches to the sea dead or alive because they are too small, have low values, or for fishery rules
Fully fished	Fish capture is at the maximum possible level; a further increase could cause overfishing
Ghost fishing	Accidental capture and killing of marine wildlife in gear, net, and traps; ghost fishing is generally lost in the sea
Introduced	Species introduced deliberately into an environment for fish farming
Maximum economic yield	Catch level allowing the most profitable fishing. It leaves “more fish in water” than Maximum Sustainable Yield
Maximum sustainable yield	Maximum average annual catch that can be removed from water: a further increase in annual catch results in overfishing
Overfishing	Fishing pressure is too high, and fish are removed at an unsustainable rate
Seafood	Any fish or shellfish intended for human consumption
Sustainable fisheries	Fishery is sustainable if the stocks of target and not-targeted species are maintained over the time. According to FAO guidelines, a sustainable fishery meets the needs of fishermen, consumers, and environment
Target species	Fish or shellfish species intended to be caught and sold

**Table 2 foods-10-02725-t002:** Classification of biorefinery according to Kamm et al. [17]; the details have been adapted from various sources.

Term	Definition
LFB (Lignocellulosic biorefinery)	Biomass largely available (straw, grass, wood, and paper-waste) The main products are cellulose, hemicellulose, and lignine
WCB (Whole Crop Biorefinery)	It uses cereals, such as rye, wheat, triticale, and maize, as input feedstock to produce ethanol or bio-plastic
GB (Green Biorefinery)	It deals with a great variety of green biomass such as grass (from cultivations, pasture lands, roadside cuttings, private gardens, and parks) and green crops (i.e., lucerne, clover, and immature cereals from intensive cultivations)
OWB (Organic Waste Biorefinery)	It uses waste generated by fish, meat, dairy, egg, vegetable, etc. Carbohydrates, proteins, and lipids represent the main recovery products generally used to produce a wide variety of compounds (bioethanol, lactic acid, succinc acid, and various enzymes such as pectinases or cellulases and hemicellulases) or biofuels.

**Table 3 foods-10-02725-t003:** Criteria and scores for the evaluation of economic and technical feasibility of a new product from by-products/discards or low-value fish (modified from Iñarra et al. [45]). Each criterion has 4 possible scores (high, medium, low, and null) reported in descending order in the table.

Macrocategory	Criterion	Scores
Case-dependent study	Available raw materials	High
		Medium
		Low
		Very low
	Available facilities	Many and/or nearby
		Far away
		Pilot plant
		Experimental
Technical	Yield of the process	High (>50%)
		Medium (10–50%)
		Low (<10%)
		Very low (0.05%)
	Technology readiness level (TRL)	High
		Medium
		Low
		Experimental
Economic	Product (value)	High
		Medium
		Low
		Null
	Market	Big (international)
		Medium (national or transnational)
		Low (Regional or local)
		None
	Costs for the production	Very low
		Low
		Medium
		High
	Competitors	None/few
		Some
		Many
		Saturated market

**Table 4 foods-10-02725-t004:** Examples of new products from low-value or discard fish.

Product	Species	Process	Product Description	Reference
Fermented surimi	Anchovy (*Engraulis* spp.), roundscad (*Decapterus* spp.), and other small pelagic and demersal species	Fermentation	Surimi blocks were prepared as mix of the reported fish species and supplemented with 2% glucose, 5% corn starch, and 1% isolated soy protein. The product was fermented by Actinomucor elegans for 36 h.	[54]
			Fermentation increased the content of histidine, phenylalanine, and glutamic acid.	
Surimi	Ungutted myctophid (*Benthosema pterotum*)	Mixing and food formulation with other ingredients	Ungutted fish was deboned, minced, and used to give structure to surimi and then mixed with onion, breadcrumbs, wheat flour, skim milk powder, sunflower oil, fresh grated garlic, and salt.	[55]
			The product had lower fish odour and flavour and better sensory scores for texture attributes than silver carp mince.	
Fish burgers fortified with algae	Common barbel (*Barbus barbus*)	Fortification	Fish mince was thawed overnight in the refrigerator and mixed thoroughly with salt (2% W/V), cornstarch (1% W/V), and different concentrations (0.5, 1, and 1.5%) of microalgae powders (Chlorella minutissima, Isochrysis galbana, Picochlorum sp.).	[56]
			The presence of microalgae gave better swelling ability and higher antioxidant levels.	
Paste and burgers made form waste or low-value fish	Red porgy (*Pagrus pagrus*), Argentine croaker (*Umbrina* sp.), Atlantic bigeye (*Priacanthus arenatus*), Black cusk-eel (*Conger* sp.), Cusk-eel (*Genypterus brasiliensis*), and Blackfin goosefish (*Lophius gastrophysus*)	Mixing	Mincing, mixing, and product designing.	[57]
Black seabream ceviche, smoked blue jack mackerel pâté, dehydrated piper gurnard, fried boarfish, and comber pastries	Blue jack mackerel (*Trachurus picturatus*), black seabream (*Spondyliosoma cantharus*), piper gurnard (*Trigla lyra*), and two unexploited species (comber, *Serranus cabrilla* and boarfish, *Capros aper*)	Mixing	Mincing, mixing, and product designing.	[49]
Packed fillet	Sea bass (*Dicentrarchus labrax*)	Biopreservation and addition of essential oils	Fillets of sea bass were inoculated with a mixture of lactic acid bacteria (Lactococcus lactis, Lactiplantibacillus plantarum, and Carnobacterium piscicola) and added with citrus essential oil and then vacuum packed.	[58]
			The product exhibited prolonged shelf-life and ameliorated muscle liquid-holding capacity.	
Thai snacks (Jeep Thai, Shor Moung, and Pun Khlip)	Nile tilapia (*Oreochromis niloticus*)	Reformulation (substitution of an ingredient)	Addition of tilapia instead of some other ingredients.	[59]
Probiotic fillets	Sea bream of the Adriatic Sea (*Sparus aurata*)	Probiotication	Marinated sea bream fillets enriched with *Lactiplantibacillus plantarum* and *Bifidobacterium animalis* subsp. *lactis*, then packed in oil or in a diluted brine.	[60]

## Data Availability

Not applicable.

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
