# Peer review of "Fish Loss/Waste and Low-Value Fish Challenges: State of Art, Advances, and Perspectives"

_foods, 2021, doi:10.3390/foods10112725_

Round 1

Reviewer 1 Report

This manuscript aims to clarify some issues related to fisheries management in term of sustainability, overfishing and fisheries wasted due to low value or  for unintentionally fished  species or sizes. The manuscript is well organized and gradually clarified and focused on the aim specified by the authors.

These are my specific comments and suggestions

I wonder why the authors do not focus on the possibility of using fish wastes to produce raw material for animal and fish feeds.  Currently global market prices for feed ingredients are rising , there is a competition for land resources to produce agricultural food for feeding the humans or the livestock. Fisheries and Aquaculture are also competing as aquaculture feeds are based on fisheries. The authors of the present work may be interested in elaborating on this issue as fish wastes could be used in this way to support the growth of the aquaculture feed industry.

I wonder if the authors can clarfy the current capacity of biorefineries (say for example in Europe) to handle fish wastes.  This would strengthen their arguments on this paper.

The figures are great but fugure legends should include some explanation of the content and the concept to help the reader. For example all legends are brief, whereas tables have a much better description of their content.

Reviewer 2 Report

Please see the comments and suggestions below

1) Introduction section needs to be restructured. Currently, it is too long (almost 2.5 pages). You need to have answered four question what is an issue, why is it an issue and how in the past/or currently this issue is being addressed and finally the structure of the paper. The rest you can create another section. 

2) Section 2 - Biorefinery - definitions and classification - Not sure why you are providing this information. This section seems irrelevant. 

3) Research methodology is missing. Follow a simple structure. Introductions, Methodology, Results and Discussions and Conclusion.

4) The paper does not seem to be novel. 

Reviewer 3 Report

The paper ‘Sustainability of Fishery also Relies on Solutions to Address Loss/Waste and Low-Value Fish Challenges: State of Art, Advances, and Perspectives’ is a trial to describe state of the art in sustainability of fishery area. Authors try to discuss the strategy of possible efficient use of unwanted ‘low-value fish’ and fish losses/wastes. The subject is important in the scope of human future and sustainable development, but the paper needs serious improvements before consideration for publication in Foods journal.

COMMENTS

The paper has to be restructured.

There is no need to add ‘biorefinery section’. The section 3 should be reformulated (e.g. ‘fish based bioactive compunds’). The section regarding ‘Chitin’ should be removed or justified as a fish based compound.

The discussion placed in section 4 doesn’t correspond to the title of the section. New products are not described.

The section ‘future perspectives’ should be improved and consists of the future possibilities, direction, findings based on the literature. Current text should be moved to discussion section

Sentences like in (l.362-363) and (l. 390-393) seems like a copy paste from other thesis or papers.

l. 375-378 – 8 solutions are listed but only one is described – all of them should be discussed in the paper

The title has to be reformulated to address the review goals like ‘Fish loss/waste and Low-Value Fish Challenges: state of Art, Advances, and Perspectives’

Rewrite abstract to make on overview of the described issues. There is no need to use description like (l. 14) ‘ after a definition …biorefinery is discussesd…’ or ‘ the paper proposes perspcetives…’ – abstract should include a short summary what aspects are discussed, what are perspectives, etc.

The paper should be carefully checked by a professional  translation office. The entire manuscript is far from the acceptable level, between others e.g. (l. 26 – ‘there are …’, word ‘are’ is missing; l. 52-54 – check if the sentence is correct; l. 60-63  – check if the sentence is correct; l. 66 – ‘a long-time fish’ – can you explain that term – is it ‘old fish’ or ‘the fish held for a long time in the net’; l. 78 – ‘that is’ should be removed; l. 81 – ‘are’ is missing, …)

MINOR

  • l. 397-398 – ‘three possible targets of consumers…’ – four groups of consumers are described
  • l. 64-78 – are these authors conclusions? Otherwise references are missing
  • l. 81-84 – are these authors conclusions? Otherwise references are missing
  • l. 100 – the definition is in table 1

Round 2

Reviewer 2 Report

The authors have done majority of the corrections and have addressed most of the comments. As your article is focussing on the sustainability of the fisheries, probably, the following article could be of interest "Co-design of food system and circular economy approaches for the development of Livestock feeds from insect larvae".   

Author Response

We thank this reviewer for his/her helpful comments. As suggested we added this possibility in the section of use of fish waste as feed for aquaculture